# OpenReview forum: "Effective and Stealthy One-Shot Jailbreaks on Deployed Mobile Vision–Language Agents"
_ICLR.cc/2026/Conference — ICLR 2026 Conference Withdrawn Submission_

### Official Review · Reviewer_4NdB · 2025-10-23

**Soundness:** 3
**Presentation:** 2
**Contribution:** 2
**Rating:** 4
**Confidence:** 3

**Summary:**

The paper introduces an attack pipeline that uses a banner pop up depicting a prompt injection to mislead a mobile agent to perform harmful actions. In order to bypass on-device safety filters the authors introduce a method for detoxifying the prompt injection text. This attack is them performed on a new data set of 40 test cases including "halmless looking" and "halmful looking" prompt injections, on the Mobile-Agent-E agent powered by 6 different VLM models.

**Strengths:**

- The paper deals with an area of critical importance and shows how known techniques can be combined to create and effective attack
- Most of the paper is well written
- The general presentation of the paper is good with well crafted figures
- Appendix sections 4 onwards are good

**Weaknesses:**

At the moment this paper lacks focus, and likely falls foul of trying to do too much at once. Leading to no one area feeling fully developed to the point of being ready for publication. The paper is not presented as a benchmark for testing environmental attacks on mobile agents and the number of test cases is limited (40), split over three apps. The attack vector presented is not substantially novel, nor is the sort of environmental injection attack used, nor is the finding that current agents are vulnerable. The paper also does not make it clear how any mobile agents would be directed to any malicious apps. The justification given is "This threat model reflects realistic scenarios where developers or maintainers could introduce malicious modifications". While I agree the developers could do this, the unanswered question is why would they? and why so more than say android developers? It seems unrealistic to me that the developers of WeChat or a SmartHome company would do this. There is no explanation of why this is more realistic than prior work. The effectiveness of the detoxification algorithm (HG-IDA*) is not compared against comparable methods of bypassing safety filters, so it is hard to judge its contribution. HG-IDA* is also poorly explained so its novelty is hard to judge.

In greater detail the weaknesses of this submission are as follows:
- An attack vector that requires just as unrealistic assumptions as prior work
- Poor baselines the related work section on jailbreak attacks has 4 papers 3 from 2 years ago, they have been many more modern jailbreaks developed in 2024, if the detoxification algorithm is a major contribution of the paper I would expect it to be compared against attacks from the last year. Moreover, section 4.3 is missing important details of how the baselines were run.
- The section on Keyword-Level Detoxification is poorly written making it hard to follow with some terms not defined.
- Lack of a clear contribution, the paper combines known ideas and hence the results do not offer much in the way of new insights.
- Confusing use of the term "zero shot". The authors claim there method is zero shot but then uses multiple queries to LlamaGuard3. To the best of my knowledge AutoDAN and GPTFuzz which rely on multi-step search could be run on against llama guard and applied "zero shot" to the target agent.

Smaller issues
- To the best of my knowledge the split of Harmful and Harmless prompt in 40 test cases is not detailed anywhere in the paper.
- Line 280 "While our previous approach using inducive prompts could disrupt the model’s alignment-based generation, harmful instruction was still blocked by content moderation." This sentence does not make sense in the context of the paper there has been no mention of "inducive prompts" or "previous approaches".
- The related work section on jailbreak attacks does not include any mention of the line of research of using adversarially perturbed images to attack multimodal agents. If the aim of this work is to present a stealthy attack, these attack which are commonly human imperceptible should be mention. I have included some reference at the bottom incase it is helpful.
- L_{max}, h() are not defined anywhere
- Missing import references best of N-jailbreaking, etc

It is clear that a lot of work has gone into this submission and with a greater focus on one area (Such as HG-IDA*) with experiments backing up the contribution I have no doubt it will form a quality publication.





Some Citations for Adversarial Image Attacks of Multimodal Agents

@article{fu2024imprompter, title={Imprompter: Tricking llm agents into improper tool use}, author={Fu, Xiaohan and Li, Shuheng and Wang, Zihan and Liu, Yihao and Gupta, Rajesh K and Berg-Kirkpatrick, Taylor and Fernandes, Earlence}, journal={arXiv preprint arXiv:2410.14923}, year={2024} }

@article{aichberger2025attacking, title={Attacking multimodal os agents with malicious image patches}, author={Aichberger, Lukas and Paren, Alasdair and Gal, Yarin and Torr, Philip and Bibi, Adel}, journal={arXiv preprint arXiv:2503.10809}, year={2025} }

@article{wu2024dissecting, title={Dissecting adversarial robustness of multimodal lm agents}, author={Wu, Chen Henry and Shah, Rishi and Koh, Jing Yu and Salakhutdinov, Ruslan and Fried, Daniel and Raghunathan, Aditi}, journal={arXiv preprint arXiv:2406.12814}, year={2024} }

@article{wang2025manipulating, title={Manipulating Multimodal Agents via Cross-Modal Prompt Injection}, author={Wang, Le and Ying, Zonghao and Zhang, Tianyuan and Liang, Siyuan and Hu, Shengshan and Zhang, Mingchuan and Liu, Aishan and Liu, Xianglong}, journal={arXiv preprint arXiv:2504.14348}, year={2025} }

**Questions:**

When you say "zero shot" in this paper what do you mean?

Are H( ) and h( ) the same thing in the Keyword-Level Detoxification?

What is "w_1"?

Why do you think it is more realistic that the developers of WeChat or a SmartHome company would deploy environmental attacks than the android developers?

Do you think HG-IDA* could be used to jailbreak LLMs?

Could you explain HG-IDA* at a high level in prose?

---

### Official Review · Reviewer_JFzR · 2025-11-03

**Soundness:** 3
**Presentation:** 2
**Contribution:** 2
**Rating:** 4
**Confidence:** 3

**Summary:**

The paper proposes a low-privilege, stealthy, and one-shot jailbreak framework that injects malicious prompts directly into app UIs. The attack relies on (1) perception-chain targeting—embedding prompts as benign-looking in-app notifications without elevated permissions; (2) user-invisible activation—revealing the prompts only during automated (ADB-driven) agent interactions; and (3) one-shot prompt efficacy—a heuristic-guided detoxification algorithm (HG-IDA*) that preserves malicious intent while evading on-device content moderation. Experiments on multiple Android apps and LVLM backends show high planning (up to 95%) and execution (up to 82%) hijack success rates. The paper highlights serious real-world risks in mobile agent deployments and proposes preliminary defense directions.

**Strengths:**

1. The proposed method achieves a substantial improvement in attack success rates compared to both baseline and ablation variants. The strong performance gain of the ensemble configuration over individual components clearly demonstrates the effectiveness and complementarity of the proposed modules.
2. The study provides actionable insights into agent design weaknesses (e.g., instruction attribution failures, cross-app pivoting) and motivates research on provenance-aware defenses.

**Weaknesses:**

1. The evaluation focuses primarily on different LVLM backends but does not examine robustness across distinct agent architectures.
2. The dataset includes only 40 scenarios, which may not capture the diversity of real-world app workflows or environmental conditions.
3. The study’s scope is relatively limited, as it focuses primarily on single-turn, text-based jailbreak attacks. It does not explore multi-turn, multi-modal, or adaptive attack settings, which could offer a more comprehensive understanding of agent robustness in real-world environments.
4. During the HG-IDA* optimization process, the papers rely on an open-source model to estimate safety scores. However, the paper does not thoroughly analyze how critical this model is to the overall attack success, nor whether the approach may overfit to the scoring model, potentially limiting generalization to an unseen domain.

**Questions:**

Could you provide a concrete example to illustrate how the prompt is modified after optimization via the HG-IDA algorithm?

---

### Official Review · Reviewer_y9qd · 2025-11-05

**Soundness:** 3
**Presentation:** 2
**Contribution:** 3
**Rating:** 4
**Confidence:** 3

**Summary:**

This paper presents a practical jailbreak framework against LVLM-based mobile agents via low-privilege, stealthy, one-shot prompt injections. The attack has three components: (1) vision-embedded malicious prompts rendered inside the app UI, (2) a trigger that distinguishes whether the operator is a human user or an automated agent to conditionally reveal the prompt, and (3) HG-IDA*, a character-level “detoxification” algorithm that rewrites prompts to bypass safety filters while preserving adversarial intent. The authors evaluate on 8 LVLM backends and 40 curated tasks, reporting high attack success rates.

**Strengths:**

* Comprehensive evaluation. Coverage spans closed models (GPT-4o, Gemini-2.0, Claude-3.5) and open models (DeepSeek-VL2, LLaVA-OneVision, Qwen-VL-Max), offering a broad view of current LVLM robustness.

* Separating thought-level success (T_asr) from execution-level success (R_asr) provides actionable insight into where defenses may be most effective.

* The cross-application attack examples convincingly show real-world impact and risk.

* Well-run experiments. The empirical setup and reporting are generally careful and clear.

**Weaknesses:**

* Trigger robustness. The operator detector hinges on ADB taps exhibiting near-zero size/pressure. Although it works in the reported setting, this heuristic appears brittle: scripted interactions can replay human-like touch traces, potentially bypassing the trigger. Overall, the entry point—distinguishing user vs. agent—feels fragile.

* While the integration is novel, the individual components lean heavily on prior ideas. The paper argues that in-app banner prompts differ from system-level pop-ups, but app developers can also show in-app pop-ups/ads without special privileges to mislead agents. Conceptually, this seems close to PopupAttack (Zhang et al., 2025), raising questions about what is fundamentally new beyond the combination and mobile focus.

Reference:
Zhang et al. Attacking Vision-Language Computer Agents via Pop-ups. ACL 2025

**Questions:**

* HG-IDA* clarity and ablation. The pruning policy is quite complex. Are chain-only pruning and atomic commit essential to performance, or could simpler variants achieve similar effectiveness/efficiency?

* Although Appendix A.7 mentions “provenance-aware prompting,” but no results are provided. What is its measured effectiveness? Have you tested other defenses, e.g., step-level audits by a separate VLM to flag harmful or misaligned actions?

---

### Official Review · Reviewer_xvVa · 2025-11-05

**Soundness:** 3
**Presentation:** 2
**Contribution:** 3
**Rating:** 2
**Confidence:** 3

**Summary:**

The paper introduces a one-shot jailbreak on mobile vision-language agents by embedding hidden in-app prompts revealed only during agent interaction. Using a character-level detoxification algorithm, it bypasses safety filters and achieves high attack success across several VLMs.

**Strengths:**

- The paper tackles a timely and well-scoped problem: the vulnerability of mobile vision–language agents to prompt injection attacks under realistic deployment conditions.
- The proposed framework is methodically designed, combining low-privilege UI-level injection, agent-triggered activation, and a character-level detoxification algorithm to evade safety filters.
- The experimental evaluation is thorough, spanning six LVLMs (both open- and closed-source) and three Android apps, and includes clear ablations and baseline comparisons.
- The evaluation setup demonstrates strong engineering and practical relevance.

**Weaknesses:**

- The conceptual novelty is limited: the work integrates known ideas (prompt injection, obfuscation, touch-based activation) rather than introducing new theoretical insights about multimodal robustness.
- The threat model assumes attacker control over app source code and additionally assumes that the agent actively uses the malicious app so that the in-app notification can be displayed. These are strong assumptions that may not generalize to typical real-world attack vectors.
- The authors claim that imperception during normal human interaction requires discriminating between agent-driven and human input (line 93). However, this is not necessarily required. For example, Aichberger et al. (2025) demonstrate that small, human-imperceptible image perturbations can reliably hijack agents without any need to detect the actor type. Such adversarial patches are deployable without elevated privileges and achieve single-attempt effectiveness against filtering mechanisms, directly contradicting the authors’ statement (line 83) that existing approaches have not yet met these criteria. These claims are therefore inaccurate and overlook relevant prior work.
- The problem formulation is very sloppy, and I would recommend rewriting the entire section. Several mathematical issues undermine its clarity and rigor:
  - The equations are not numbered, making it difficult to reference them.
  - The optimization objective (line 213) does not rigorously include or depend on the safety term, even though it is introduced as a key part of the formulation.
  - The formalization conflates agent-driven and human-driven interactions. The state definition $s't$ depends on the binary trigger function, which equals 1 only when the agent acts. Yet the objective is written without conditioning on this case.
  - The authors define $\mathcal{A}$ as the agent’s action space but later write $\mathcal{A}(G_a)$. Since $\mathcal{A}$ is a set, not a function, this notation is incorrect.
  - Similarly, $s_t$ is defined as a visual state and $\delta$ as a text sequence. These belong to different modalities, so expressions like $s'_t = s_t + \delta$ are undefined.
- The presentation is overly detailed at the expense of conceptual clarity and broader implications (to give one example, Section 3.4 largely repeats Section 3.2 and disrupts the logical flow).

---
Aichberger, L., Paren, A., Torr, P., Gal, Y., & Bibi, A. (2025). Attacking Multimodal OS Agents with Malicious Image Patches. arXiv preprint arXiv:2503.10809.

**Questions:**

- How realistic is the threat model: how often can an attacker realistically modify an app’s source without elevated privileges, and does that limit real-world impact?
- How were the injected toasts actually rendered on the device? The paper claims they were styled identically to benign toasts, yet Figure 1 shows a visually unpolished and very large banner.
- How dependent is the attack on ADB-driven taps: does it work with other agent control channels or when touch-pressure/size signals are noisy?
- How robust are the results? Do $T_{asr}$ and $R_{asr}$ success rates hold under small UI/timing variations and across repeated runs, or are successes brittle and hard to reproduce?

---

### Note · Authors · 2025-11-20

I have read and agree with the venue's withdrawal policy on behalf of myself and my co-authors.